# Evaluation of Thrombomodulin, hsa-miR-18a-5p, and hsa-miR-18b-5p as Potential Prognostic Biomarkers in Uterine Corpus Endometrial Carcinoma

**DOI:** 10.3390/ijms26083649

**Published:** 2025-04-12

**Authors:** Enes Karaman, Ergul Bayram, Durmus Ayan

**Affiliations:** 1Department of Obstetrics and Gynecology, Faculty of Medicine, Nigde Omer Halisdemir University, 51240 Nigde, Turkey; 2Medical Biochemistry, Nigde Omer Halisdemir University Research and Training Hospital, 51100 Nigde, Turkey; 3Department of Medical Biochemistry, Faculty of Medicine, Nigde Omer Halisdemir University, 51240 Nigde, Turkey

**Keywords:** uterine corpus endometrial carcinoma, thrombomodulin, hsa-miR-18a-5p, hsa-miR-18b-5p

## Abstract

Thrombomodulin (THBD), hsa-miR-18a-5p, and hsa-miR-18b-5p have been frequently mentioned in numerous cancer-related research studies; however, their specific functions in uterine corpus endometrial carcinoma (UCEC) are not well understood. This study aimed to investigate the roles of THBD, hsa-miR-18a-5p, and hsa-miR-18b-5p within a UCEC cohort. We utilized various web-based tools, including GEPIA2, UALCAN, Human Protein Atlas (HPA), TNM Plot, STRING, TargetScan, and ENCORI for our analysis. The expression level of the THBD gene was found to be significantly downregulated (*p* < 0.05) in UCEC tissue compared to adjacent normal tissue. In contrast, hsa-miR-18a-5p and hsa-miR-18b-5p were both upregulated in UCEC tissue (*p* < 0.05). Additionally, THBD exhibited a significant hypermethylation level in UCEC tissue (*p* < 0.05). The elevated expression of hsa-miR-18a-5p was linked to a shorter overall survival (OS) (*p* = 0.025), while THBD and hsa-miR-18b-5p showed no association with OS (*p* = 0.87 and *p* = 0.56, respectively). Notably, THBD expression was significantly negatively correlated with hsa-miR-18a-5p (*p* = 0.00407), whereas no significant correlation was found between THBD and hsa-miR-18b-5p (*p* = 0.25). Thus, it can be concluded that increased levels of miR-18a-5p in the UCEC cohort may serve as a negative prognostic marker and a potential therapeutic target. However, further studies are necessary to validate the implications of decreased THBD and increased miR-18b-5p expression levels on the clinical outcomes of these patients.

## 1. Introduction

Endometrial cancer (EC) is the second most frequently diagnosed gynecological cancer, accounting for approximately 4.4% of all cancer cases worldwide [1,2]. Although a significant 90% of women with EC report postmenopausal bleeding (PMB), only around 10% of PMB occurrences are actually due to EC [1]. This symptom often leads to early diagnosis, which contributes to a relatively high survival rate for patients with early-stage disease [3]. In contrast, advanced EC presents substantial treatment challenges and requires a multidisciplinary strategy that includes surgery, radiotherapy, and chemotherapy. The prognosis for patients can vary greatly based on the stage at diagnosis, the histological subtype, and the molecular characteristics of the cancer [4]. Current diagnostic approaches, including transvaginal ultrasound, hysteroscopy, and endometrial biopsy, are invasive and costly, with limitations in specificity, highlighting the necessity for improved strategies for early detection [5].

Recent research has investigated the molecular foundations of Uterine Corpus Endometrial Carcinoma (UCEC), uncovering genetic and molecular factors that contribute to its development [1,6]. Among these factors is the thrombomodulin (THBD) gene, which encodes a glycoprotein present on endothelial cells that plays a crucial role in regulating coagulation through the activation of protein C, which has anticoagulant, anti-inflammatory, and cytoprotective functions [7,8,9]. Beyond its conventional role in blood clotting, THBD has been linked to various cellular processes, including inflammation, cell proliferation, and angiogenesis, all of which are significant in the context of cancer biology [10]. Emerging data suggest that THBD, along with miR-18a-5p and miR-18b-5p, may be involved in the pathophysiology of UCEC [8].

The presence of atypical thrombomodulin expression in tumor tissues has been documented, raising important questions about its function in tumorigenesis, progression, and metastasis. Certain evidence implies that changes in the expression or activity of thrombomodulin may be associated with the prothrombotic state often observed in cancer patients, which is related to worse clinical outcomes [11,12,13]. Moreover, tumor type, treatment, and patient-specific factors all contribute to cancer-associated venous thromboembolism (VTE), but their individual roles and underlying mechanisms remain unclear. There are also data suggesting that chemotherapy alters serum thrombomodulin levels and confers acquired activated protein C (aPC) resistance, which may play a role in cancer-related VTE in gynecologic cancer patients [14].

A comprehensive understanding of the interactions between THBD, related miRNAs, and UCEC could lead to significant revelations about the molecular mechanisms of the disease and highlight potential therapeutic targets. Therefore, this study is designed to investigate the relationships among THBD, hsa-miR-18a-5p, and hsa-miR-18b-5p in the context of UCEC. As far as we know, no previous studies have examined THBD, hsa-miR-18a-5p, and hsa-miR-18b-5p together within the UCEC cohort, particularly in relation to overall survival, gene expression levels, promoter methylation status, and protein interactions. In this regard, our research is pioneering.

## 2. Results

### 2.1. The Expression Profile of THBD, hsa-miR-18a-5p and hsa-miR-18b-5p

The expression of THBD was assessed using the Human Protein Atlas (HPA) (https://www.proteinatlas.org/, accessed on 11 January 2025). The staining of tumor cells with the HPA002982 (Anti-THBD antibody) revealed a medium intensity (Figure 1A). According to the GEPIA2 database, THBD expression was found to be significantly reduced in UCEC tissue when compared to normal adjacent tissue (*p* < 0.05) (Figure 1B). Furthermore, no significant differences were observed across the various stages of UCEC (Figure 1C). Lastly, data from the ENCORI database indicated that the expression levels of hsa-miR-18a-5p and hsa-miR-18b-5p were significantly elevated in UCEC tissue relative to normal adjacent tissue (*p* < 0.05) (Figure 1D).

### 2.2. The Promotor Methylation Status of THBD

The level of THBD methylation was found to be significantly elevated in UCEC tissue (*n* = 438) when compared to the healthy adjacent tissue (*n* = 46), with a statistical significance of *p* < 0.05 (Figure 2).

The expression of THBD in UCEC demonstrated a statistically significant downregulation when comparing normal tissue to each cancer stage, including stage I, stage II, and stage III (*p* < 0.05). In terms of racial demographics, THBD expression was also significantly lower in normal tissue compared to Caucasian, African-American, and Asian patients (*p* < 0.05). Additionally, when considering patient weight, THBD expression was statistically diminished in comparisons of normal versus normal weight, normal versus extreme weight, normal versus obese, and normal versus extremely obese individuals (*p* < 0.05). Regarding menopause status, a significant downregulation of THBD expression was noted in normal versus pre-menopausal and normal versus post-menopausal groups (*p* < 0.05). Finally, the analysis of histological subtypes revealed a statistically significant reduction in THBD expression in normal versus endometrioid, normal versus serous, normal versus mixed serous and endometrioid, as well as between endometrioid versus serous and endometrioid versus mixed serous and endometrioid (*p* < 0.05). (Figure 3).

### 2.3. The TNMplot and muTarget Analysis of THBD

The expression of the THBD gene exhibited statistically significant alterations across all cancer types when comparing tumor tissue to normal adjacent tissue (*p* < 0.05), with the exception of bladder cancer, which showed no significant difference (*p* > 0.05) (Figure 4).

The expression of THBD was found to be statistically associated with mutations in TP53, NCAPD3, CFAP54, FBXL7, PDE8A, and ADAM22 (*p* < 0.05), as shown in Figure 5.

### 2.4. The Survival Analysis Results of THBD, hsa-miR-18a-5p and hsa-miR-18b-5p

The analysis revealed that the expression levels of THBG (*p* = 0.87) and hsa-miR-18b-5p (*p* = 0.56) did not exhibit a statistically significant relationship with overall survival (OS). However, a significant association was found between decreased levels of hsa-miR-18a-5p (*p* = 0.025) and prolonged OS in UCEC, as shown in Figure 6.

### 2.5. The Gene–Gene Interaction Results

An illustration of a protein network derived from the STRING database, focusing on the THBD gene, is presented in Figure 7. The hues of the lines that link the protein nodes indicate the type of evidence that underpins their functional relationship, whereas the spacing between the nodes signifies the confidence level of the interaction, as assessed by a Bayesian scoring methodology (Table 1).

### 2.6. The Correlation Results Between THBD with hsa-miR-18a-5p and hsa-miR-18b-5p

Based on the data acquired through the TargetScan 8.0 web tool, the two conserved miRNAs most closely associated with the THBD gene are hsa-miR-18a-5p and hsa-miR-18b-5p.

A statistically significant negative correlation was observed between the expression level of THBD and hsa-miR-18a-5p (*p* = 0.00407), whereas no significant correlation was detected between THBD and hsa-miR-18b-5p (*p* = 0.25) (Figure 8).

### 2.7. DEGs Analysis Result

In the GSE7305 dataset, the THBD gene (ID: 203887) was statistically significantly upregulated in UCEC compared to the control (adjusted *p* value = 0.00105, fold-change = 1.45) (Figure 9A). In the GSE25628 dataset, the THBD gene (ID: 203888) was statistically significantly upregulated in UCEC compared to control (adjusted *p* value = 0.0467, fold-change = 1.25) (Figure 9B). THBD gene expression was found to be significantly and moderately elevated in UCEC in both databases (GSE7305 and GSE25628).

## 3. Discussion

This study investigated the expression of THBD, hsa-miR-18a-5p, and hsa-miR-18b-5p in the UCEC cohort. A significant finding was the downregulation of THBD gene expression, while hsa-miR-18a-5p and hsa-miR-18b-5p exhibited increased levels. Notably, we also discovered that higher levels of hsa-miR-18a-5p were associated with reduced overall survival (OS). To our knowledge, this is the first study to perform a bioinformatics analysis of THBD, hsa-miR-18a-5p, and hsa-miR-18b-5p in conjunction with the UCEC cohort.

This investigation revealed that THBD expression is significantly diminished in cancer cells, while miR-18a-5p levels are notably increased when compared to normal endometrial cells. The heightened levels of miR-18a-5p were associated with enhanced cancer cell proliferation, migration, and invasion, alongside a reduction in THBD activity, as demonstrated through CCK-8, Transwell, and dual Luciferase assays. These findings suggest that miR-18a-5p may inhibit THBD, thereby facilitating cancer progression. This regulatory pathway could potentially be targeted for the development of new cancer therapies aimed at preventing tumor growth and metastasis [8].

A different study has shown that a decrease in thrombomodulin expression is associated with increased tumor cell invasion and a poorer prognosis in non-small cell lung cancer [12]. In our analysis of the UCEC cohort, we aimed to explore the relationship between THBD expression and prognosis, focusing exclusively on overall survival (OS). However, we found no evidence of a correlation between lower THBD gene expression levels and OS in this cohort. This outcome may be attributed to several potential factors.

UCEC presents as a complex disease with various subtypes, each possessing unique genetic and molecular characteristics. Prognosis is significantly influenced by factors such as hormone receptor status (including estrogen and progesterone), mutations in the TP53 gene, microsatellite instability, and other genomic alterations [15]. As a result, the impact of a single gene like thrombomodulin may be less significant when compared to other more critical factors that affect overall survival (OS). Analysis using muTarget has shown that thrombomodulin levels are lower in cases with mutations in TP53, NCAPD3, CFAP54, FBXL7, PDE8A, and ADAM22 compared to those without these mutations. This suggests that the simultaneous presence of these mutations may obscure the relationship between thrombomodulin and OS. muTarget analysis includes the collective analysis results of different mutations such as single nucleotide polymorphisms (SNPs) and de-nova mutations. The role of which type and which region of mutation on Thrombomodulin is unknown. It is clear that more comprehensive experimental studies are needed to determine the mutations that affect Thrombomodulin levels. However, we believe that our study can lead to new studies in terms of revealing the deficiency in this research area. Thrombomodulin is primarily known for its anticoagulant role, acting as a cofactor in the activation of protein C by thrombin, and it also contributes to inflammation, cell proliferation, and vascular homeostasis [16]. However, its influence on cancer biology, particularly in UCEC, may not directly affect cancer progression, metastasis, or patient survival.

Alterations in gene expression do not always result in functional changes. For example, lower levels of thrombomodulin gene expression may not have a substantial effect on its protein function or the downstream pathways in UCEC. Compensatory mechanisms or alternative pathways may counterbalance the effects of reduced thrombomodulin, making its gene expression less significant for overall survival. Thrombomodulin is not typically categorized as an oncogene or tumor suppressor gene, in contrast to other genes that are directly involved in cancer progression. Its main roles pertain to endothelial and vascular biology, which may clarify the lack of a strong association between its levels and overall survival in UCEC. Furthermore, numerous clinical factors—such as age, cancer stage at diagnosis, comorbidities, treatment strategies, and therapeutic responses—can have a more substantial effect on survival outcomes, potentially obscuring any connection between thrombomodulin levels and overall survival.

The findings of the current study align with existing literature; however, there is a lack of definitive information regarding the influence of miR-18a-5p and THBD on OS. Literature indicates that increased expression of miR-18a-5p leads to the suppression of THBD gene expression. Consequently, the observed reduction in THBD levels within our UCEC cohort may be attributed to heightened levels of miR-18a-5p. The correlation analysis revealed an inverse relationship between miR-18a-5p and THBD. Nonetheless, the only variable associated with reduced OS was elevated expression of miR-18a-5p. Although THBD and miR-18a-5p are inversely related, their roles in cancer biology and survival differ significantly. miR-18a-5p has a direct oncogenic effect that contributes to decreased survival, while THBD’s impact on survival appears to be minimal or secondary. This distinction may clarify why only miR-18a-5p shows a significant association with overall survival in UCEC.

These results suggest that increased levels of miR-18a-5p in our UCEC cohort are indicative of a poor prognosis. Furthermore, hypermethylation in the THBD promoter region was noted in this cohort, which may have contributed to the downregulation of THBD by silencing the gene. A related study has found that diminished levels of miR-18b-5p in ovarian cancer cells target VMA21, thereby inhibiting both proliferation and metastasis [16].

According to findings from a study on lung adenocarcinoma, the ZFPM2-AS1 molecule was shown to lower the levels of miR-18b-5p, which subsequently leads to an increase in VMA21 levels [17]. This rise in VMA21 is associated with the promotion of lung adenocarcinoma growth. Both studies sought to clarify the role of miR-18b-5p in cancer progression [18]. Conversely, another study indicated that LncRNA AC073284.4 can inhibit epithelial-mesenchymal transition (EMT) and reduce migration in breast cancer cells by influencing the miR-18b-5p/DOCK4 axis [19].

The dualistic nature of miR-18b-5p’s effects on various molecules in different cancer types complicates its role in cancer research. In our UCEC cohort, we identified a statistically significant upregulation of miR-18b-5p levels relative to healthy tissue. However, this increase did not demonstrate a statistical association with overall survival (OS). Within our UCEC cohort, OS was the only prognostic measure available for evaluating miR-18b-5p levels. It is particularly interesting that miR-18b-5p levels show distinct characteristics across different molecules. Clarifying these pathways could provide valuable insights into cancer pathogenesis and aid in the development of innovative and effective treatment options.

The understanding of how the thrombomodulin gene interacts with other genes and the pathways it participates in is significantly influenced by the scores obtained from gene–gene relationships through string analysis. For example, thrombomodulin carries out some of its functions via the Endothelial protein C receptor (PROCR). This receptor, located on endothelial cells, plays a crucial role in enhancing protein C activation through the thrombin–thrombomodulin complex, thus regulating the protein C pathway to prevent organ damage from various stressors [20]. According to the string analysis scores, the gene that interacts most closely with thrombomodulin is the PROCR gene. Additionally, PROCR is reported to be vital for cell division, apoptosis, proliferation, and tumor recurrence. To fully elucidate the impact of thrombomodulin within the UCEC cohort, it is essential to explore the primary pathways related to other interacting genes, as suggested by the scoring data [16].

The analyses performed with the UALCAN database revealed critical information regarding the fluctuations in THBD gene expression in relation to various factors, including patient age, cancer stage, race, menopausal status, weight, and histological subtypes. Uterine corpus endometrial carcinoma (UCEC) is often perceived as having a favorable prognosis in its early stages (stage I and stage II, according to the 2009 International Federation of Gynecology and Obstetrics classification [21]. The five-year survival rates for patients diagnosed with stage I or II endometrial carcinoma typically fall between 82% and 90%. However, for those diagnosed at stage III or IV, the survival rates significantly drop to between 34% and 42%. Thus, accurate initial diagnosis and timely intervention are vital for the effective management of endometrial carcinoma.

The early identification of UCEC allows for the implementation of less invasive surgical options, which in turn reduces the requirement for additional treatments. This method not only lowers healthcare costs but also diminishes morbidity and mortality rates. Currently, there is a lack of a highly accurate and reliable early detection test for UCEC that can be effectively applied to women at high risk or those suspected of having the condition [22]. Although the significant reduction in THBD gene expression when compared to normal tissue, particularly in stage III UCEC, is encouraging for diagnosing advanced stages, the absence of changes in THBD expression across different stages limits its effectiveness as a marker. The majority of endometrial cancer patients are postmenopausal, with an average age of diagnosis being 60 years [23]. In the UCEC cohort we studied, the notably lower THBD gene expression levels in the 61–80 age group compared to the 21–40 and 41–60 age groups may suggest that THBD is a significant biomarker for identifying cancer at younger ages.

A notable reduction in THBD gene expression levels was observed in comparison to normal tissue, particularly among various histological subtypes. The most pronounced decline was evident in the serous, mixed-serous, and endometrioid subtypes. Patients diagnosed with serous subtype tumors tend to have a diminished 5-year survival rate, attributed to a greater propensity for metastasis and an elevated risk of recurrence. Consequently, the identification of tumor subtypes is crucial for formulating personalized treatment strategies and assessing prognosis [24].

The observed decline in THBD gene expression in the serous subtype, in contrast to the endometrioid subtype, could play a significant role in tailoring treatment and assessing prognosis for patients diagnosed with the serous subtype. Research suggests that miR-18a-5p may enhance the proliferative, migratory, and invasive characteristics of endometrial cancer cells, while THBD appears to mitigate these traits [8]. Furthermore, findings from the same study indicate that miR-18a-5p negatively influences THBD expression, and that the oncogenic impact of miR-18a-5p on endometrial cancer cells is counteracted by the overexpression of THBD. In our analysis of the UCEC cohort, we observed elevated levels of miR-18a-5p alongside reduced levels of THBD. Notably, THBD expression was lower in the serous subtype compared to both normal tissue and the endometrioid subtype, potentially due to the inhibitory effect of miR-18a-5p on THBD. The observed decrease in THBD and increase in miR-18a-5p may contribute to the development of the serous subtype by promoting invasion. Given the heightened risk of metastasis and recurrence associated with the serous subtype, the levels of both miR-18a-5p and THBD could serve as valuable biomarkers for tracking the aggressive progression of cancer.

Ejlalidiz et al. listed the genes they found to be associated with UCEC in their bioinformatic study in 2024 using GEPIA2 database software (http://gepia2.cancer-pku.cn/, accessed on 11 January 2025). The genes that were interacting were divided into 7 modules, and THBD was included in the 6th module. In our study, it was determined by the analysis that THBD was moderately and significantly high in UCEC. In order to determine the exact role of this gene in UCEC, the mutations it is associated with, the proteins and genes it is associated with, should be clarified with experimental studies. Our study has the potential to create preliminary data for other studies [25].

Various clinical factors, including age, cancer stage, comorbidities, and treatment response, significantly influence survival outcomes and may obscure the link between thrombomodulin levels and OS. UCEC primarily affects postmenopausal women due to estrogen fluctuations. While early-stage cases have a high 5-year OS rate (>90%), advanced or recurrent UCEC has a poor prognosis (<30%). Risk factors such as smoking, alcohol use, obesity, and hypertension contribute to its development [26]. In our UCEC cohort, the lowest thrombomodulin levels were observed in the postmenopausal period, while a decrease in thrombomodulin levels was observed with age. Thrombomodulin may be effective in the development of UCEC in addition to all these clinical factors. On the other hand, there was no statistical difference in THBD gene expression between extreme weight, obese and extremely obese groups in the UCEC cohort.

The results highlight the potential of the miR-18a-5p/THBD axis as a novel target for therapeutic intervention. Inhibiting miR-18a-5p or restoring THBD levels may be crucial in the context of cancer development. Future therapeutic approaches could incorporate miRNA inhibitors or gene therapies that aim to elevate THBD expression, thus providing a targeted strategy to mitigate cancer progression.

## 4. Limitation

Despite the strong design of our study, it is important to acknowledge certain limitations. The most notable limitation was the disparity in the number of tumor samples compared to normal adjacent tissue samples, especially concerning the analyses of promoter region methylation and gene expression. Given that these data were obtained from the TCGA database, we faced restrictions regarding the sample sizes available, which limited our ability to enhance these numbers. This limitation may influence the reliability of our statistical comparisons. Consequently, these factors could affect the generalizability and statistical power of our results. Future research that incorporates larger and more evenly distributed sample sizes, along with access to comprehensive clinical data, would be advantageous for validating our findings and further clarifying the role of the miR-18a-5p/THBD pathway in UCEC.

## 5. Materials and Methods

This study is based on bioinformatics methodologies. In our evaluation of the prognostic features associated with THBD, hsa-miR-18a-5p, and hsa-miR-18b-5p, we employed various bioinformatic databases.

### 5.1. The Expression Profile Analysis of THBD

The expression profile analysis was conducted using the GEPIA2 database (http://gepia2.cancer-pku.cn/, accessed on 11 January 2025). GEPIA2 is an enhanced version of the web server that provides detailed gene expression analysis, including the quantification of transcription levels, analysis of particular cancer subtypes, and the option for users to upload their RNA-seq fragments for comparison with existing datasets [27]. This platform allows for the comparison of gene expression levels in tumor tissues versus adjacent healthy tissues. It also facilitates the examination of gene expression across different stages of cancer, depending on the type of cancer. In our analysis of the UCEC cohort, we compared the expression levels of THBD in adjacent healthy tissue (*n* = 91) with those in tumor tissue (*n* = 174).

### 5.2. The Methylation Status and Survival Analysis of THBD

UALCAN (https://ualcan.path.uab.edu/, accessed on 11 January 2025) serves as a comprehensive and user-friendly interactive platform designed for the analysis of cancer transcriptomic data. Its primary objective is to enhance the exploration of data from The Cancer Genome Atlas (TCGA), allowing researchers to investigate gene expression profiles and conduct detailed analyses across various cancer types. UALCAN also offers insights into the methylation status of gene promoters, enabling users to compare methylation levels between tumor and normal samples. This server is instrumental in identifying potential gene interactions and pathways [28]. We utilized the UALCAN web server to conduct promoter region methylation analyses on tumor tissues (*n* = 438) and normal tissues (*n* = 46), while also examining the expression profiles of additional input genes in Uterine Endometrial Carcinoma (UCEC), including THBD. Furthermore, we explored the variations in THBD gene expression in relation to patient age, cancer stage, race, menopause status, weight, and histological subtypes using UALCAN.5.3. The Correlation and muTarget analysis of THBD.

### 5.3. The Correlation and muTarget Analysis of THBD

The TNMplot database provides users with the capability to analyze gene expression variations in real-time across tumor, normal, and metastatic tissues for all genes, utilizing multiple platforms. The analysis portal is available without the need for registration at www.tnmplot.com (accessed on 11 January 2025) and presents three unique analysis options. Among these is the pan-cancer analysis tool, which facilitates the simultaneous comparison of normal and tumor samples across 22 distinct tissue types [29]. Additionally, muTarget (https://www.mutarget.com/analysis/, accessed on 11 January 2025) is an online resource designed to forecast the correlation between genetic mutations and alterations in gene expression [30]. We used this web tool to examine the relationship between the most common somatic mutations occurring in UCEC and *THBD* gene expression.

### 5.4. The Gene–Gene Interaction

In this study, we leveraged the STRING database (https://string-db.org/, accessed on 11 January 2025) to explore the interactions and possible mechanisms of *THBD* with other related proteins. The STRING database systematically gathers both physical interactions and functional relationships, facilitating the integration of protein–protein interactions. Data are obtained from multiple sources, including automated text mining of scientific literature, computational predictions from co-expression, conserved genomic contexts, and interaction experiment databases. Each interaction is critically analyzed, scored, and then automatically mapped to less-studied organisms using hierarchical orthology information [31].

### 5.5. MicroRNA Target Analysis

We applied TargetScan 8.0 (https://www.targetscan.org/vert_80/, accessed on 11 January 2025) to discover potential target genes, focusing on predicting the target genes of miRNAs that are differentially expressed [32,33].

### 5.6. The Relationship of MicroRNA vs. UCEC and MicroRNA vs. RNA

The ENCORI Pan-Cancer Analysis Platform (https://rnasysu.com/encori/index.php, accessed on 11 January 2025) has been designed to elucidate Pan-Cancer Networks that encompass lncRNAs, miRNAs, pseudogenes, snoRNAs, RNA-binding proteins (RBPs), and all protein-coding genes. This platform accomplishes its objectives by examining the expression profiles of these elements across 32 different cancer types, drawing on data from around 10,000 RNA-seq and 9900 miRNA-seq samples sourced from the TCGA project [34].

### 5.7. Detection of Differentially Expressed Genes (DEGs)

The Gene Expression Omnibus (GEO) DataSets were used in this study for evaluating differential expressions of THBD. The datasets analyzed were GSE7305 and GSE25628 from the GEO database. Affymetrix human U133 plus 2.0 array, was used to transcriptionally profile both normal and diseased endometrial human tissues [35,36]. Analyses were conducted using the GEO2R tool (https://www.ncbi.nlm.nih.gov/geo/geo2r/, accessed on 9 January 2025) to identify differentially expressed genes (DEGs) in both datasets. This tool operates using GEOquery (https://bioconductor.org/packages/release/bioc/html/GEOquery.html, accessed on 9 January 2025) and limma (https://bioconductor.org/packages/release/bioc/html/limma.html, accessed on 9 January 2025) to process microarray data and detect DEGs. In this study, multiple testing corrections were applied using the Benjamini and Hochberg false discovery rate method to compute the adjusted *p*-values. A log2 fold change threshold of 1 was used, and the significance level for the adjusted *p*-value was maintained at 0.05 by default. Genes with an adjusted *p*-value below 0.05 and Log2(FC) < −1 were classified as downregulated, while those with an adjusted *p*-value below 0.05 and Log2(FC) > 1 were identified as upregulated [37].

### 5.8. Statement of Ethics

The research data were derived from publicly available resources, including the TCGA database, thus negating the requirement for ethical approval.

## 6. Conclusions

In conclusion, this study provides compelling evidence that elevated expression of hsa-miR-18a-5p in UCEC is significantly associated with adverse clinical outcomes, specifically a reduction in OS. Our integrated bioinformatics and experimental analyses suggest that miR-18a-5p may function as a key oncogenic regulator in UCEC pathogenesis by promoting tumor cell proliferation, migration, and invasion, in part through the suppression of THBD expression. This inverse relationship between miR-18a-5p and THBD expression supports the existence of a potentially critical miR-18a-5p/THBD regulatory axis, which may drive tumor aggressiveness, particularly in the more malignant histological subtypes such as serous carcinoma.

While the role of miR-18a-5p as both a therapeutic target and a negative prognostic biomarker is reinforced by our findings, the clinical significance of miR-18b-5p and THBD expression levels remains less conclusive. Despite their altered expression patterns, we did not observe a statistically significant association with OS for either miR-18b-5p or THBD in the UCEC cohort. This may be attributed to the complex molecular heterogeneity of UCEC, the influence of concurrent genomic mutations (e.g., TP53, NCAPD3, CFAP54, FBXL7, PDE8A, and ADAM22), and the interplay of compensatory signaling pathways that may obscure the direct prognostic relevance of these markers.

Moreover, the observed hypermethylation in the THBD promoter region suggests that epigenetic silencing could further contribute to the downregulation of THBD expression, adding another layer of complexity to the regulation of this gene in endometrial cancer. Our correlation analyses and supporting literature indicate that the oncogenic activity of miR-18a-5p may be partially mediated by its capacity to downregulate THBD, which otherwise may exert a tumor-suppressive function through its involvement in vascular homeostasis, inflammation, and cell proliferation control.

Given the significant findings regarding the miR-18a-5p/THBD interaction, future research should prioritize functional validation through in vitro and in vivo models, explore epigenetic regulatory mechanisms, and assess the utility of these biomarkers across diverse clinical subgroups and stages of UCEC. Additionally, expanding the scope of the investigation to include other endpoints beyond OS, such as progression-free survival, recurrence rates, and treatment responsiveness, could offer more nuanced insights into their prognostic value.

Ultimately, this study lays the groundwork for the development of personalized therapeutic strategies aimed at modulating miR-18a-5p expression or restoring THBD activity. Approaches such as miRNA-based therapies (e.g., miR-18a-5p inhibitors), epigenetic modifiers, or *THBD* gene replacement techniques may represent promising avenues to counteract tumor progression and improve clinical outcomes in UCEC patients. Continued exploration of this axis holds the potential to enhance early detection, refine prognostic tools, and deliver targeted, effective treatments for individuals affected by this malignancy.

## Figures and Tables

**Figure 1 ijms-26-03649-f001:**
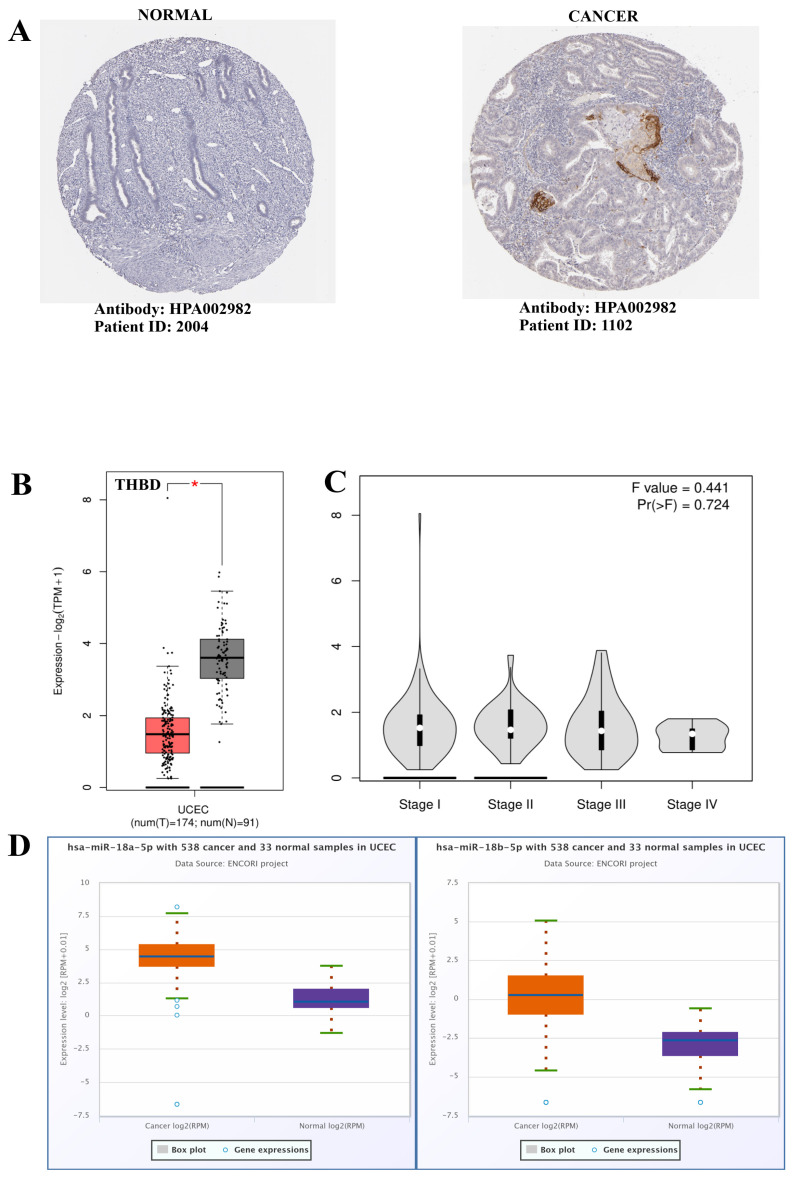
(**A**) Immunohistochemical staining images for validation of THBD expression with Human Protein Atlas (HPA), (**B**) Box plot image for the expression of *THBD*, (**C**) Violin plot image for the expression of *THBD* according to stages of UCEC, (**D**) Box plot image for the expression of *hsa-miR-18a-5p* and *hsa-miR-18b-5p*. Asterisks (*) indicate statistically significant differences (*p* < 0.05).

**Figure 2 ijms-26-03649-f002:**
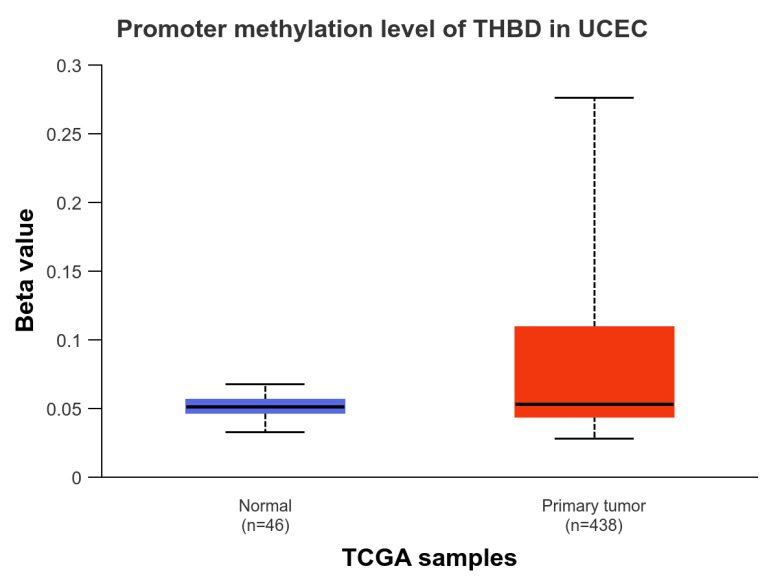
Box plot image for the promotor methylation of THBD. The Beta value indicates the level of DNA methylation ranging from 0 (unmethylated) to 1 (fully methylated).

**Figure 3 ijms-26-03649-f003:**
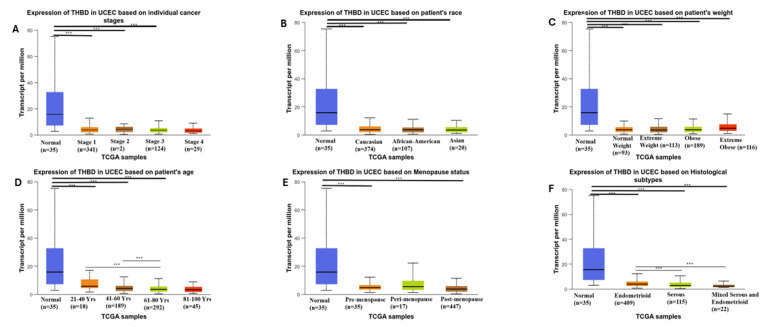
Box plot representations of THBD gene expression in UCEC based on various clinical parameters: (**A**) Cancer stages; (**B**) Patient race; (**C**) Patient weight; (**D**) Patient age; (**E**) Menopausal status; (**F**) Histological subtypes. Asterisks (***) indicate statistically significant differences (*p* < 0.05). Normal Weight: Body Mass Index (BMI) is greater than equal to 18.5 and BMI is less than 25, Extreme Weight: BMI is greater than equal to 25 and BMI is less than 30, Obese: BMI is greater than equal to 30 and BMI is less than 40, Extreme Obese is greater than 40.

**Figure 4 ijms-26-03649-f004:**
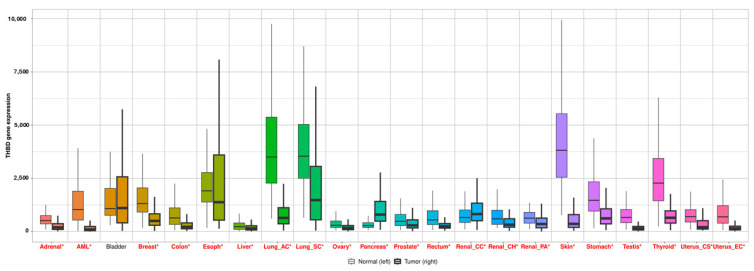
Box plots illustrate differential expression levels of THBD across various cancer types, comparing tumor and adjacent normal tissues. (AML: akut myeloid leukemia, Lung_AC: Lung Adenocarcinoma, Lung_SC: Lung Sarcoidcarcinoma, Renal_ CC: renal cell carcinoma, Renal_ CH: Renal Chromophobe, Renal_PA: Renal Papillary Cell Carcinoma, Uterus_CS: Uterine Carcinosarcoma, Uterus_EC: Uterine Corpus Endometrial Carcinoma). Red labels denote statistically significant differences (*p* < 0.05, Mann–Whitney U test). Asterisks (*) indicate statistically significant differences (*p* < 0.05).

**Figure 5 ijms-26-03649-f005:**
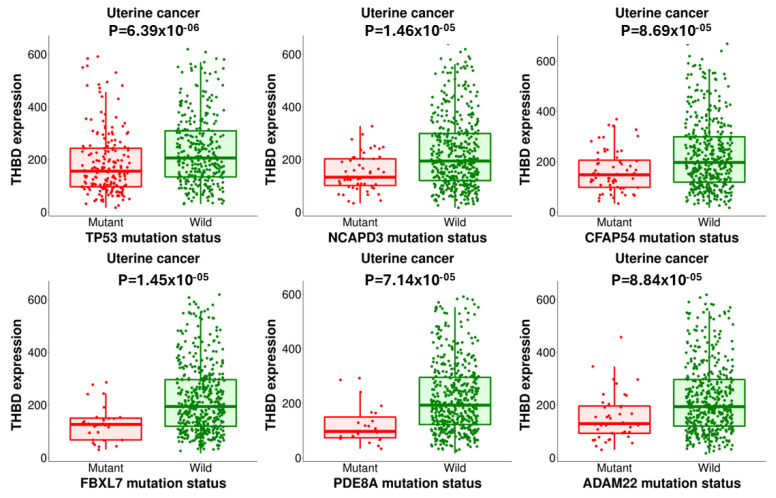
Boxplot for expression changes in THBD gene accompanied by somatic mutations. The red boxplot represents the expression change in the THBD gene with the somatic mutation type. The green boxplot represents the expression change in the THBD gene with the mild type gene present.

**Figure 6 ijms-26-03649-f006:**
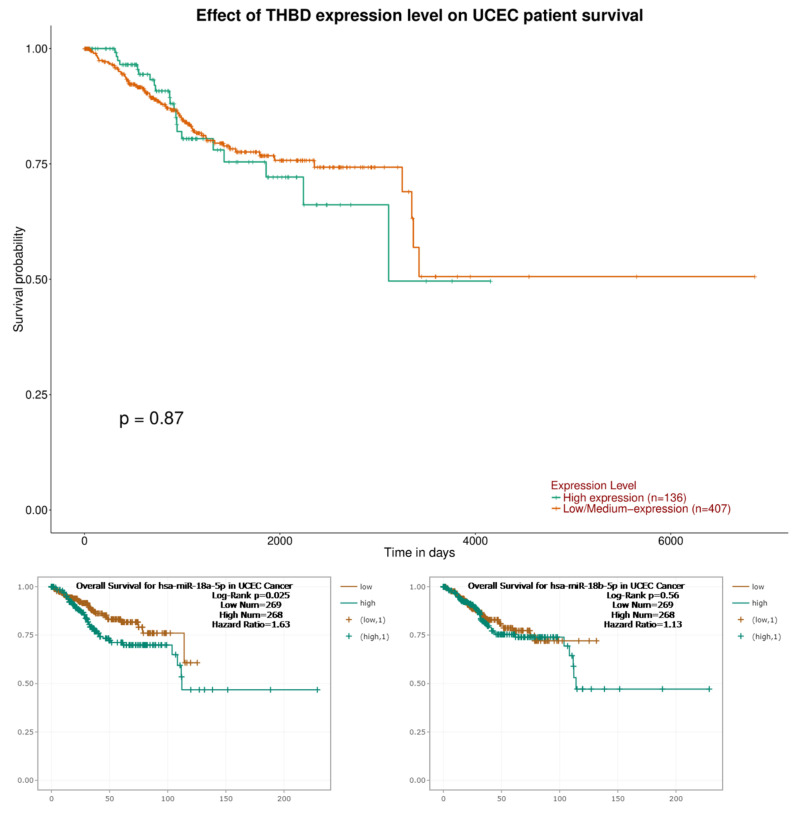
Km plotter for the relationship between OS and the expression level of *THBD*, *hsa-miR-18a-5p* and *hsa-miR-18b-5p* in UCEC.

**Figure 7 ijms-26-03649-f007:**
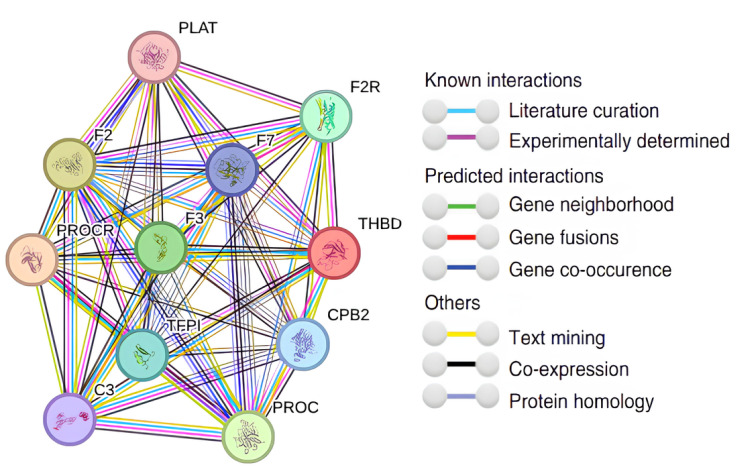
The string analysis for gene–gene interactions for *THBD*.

**Figure 8 ijms-26-03649-f008:**
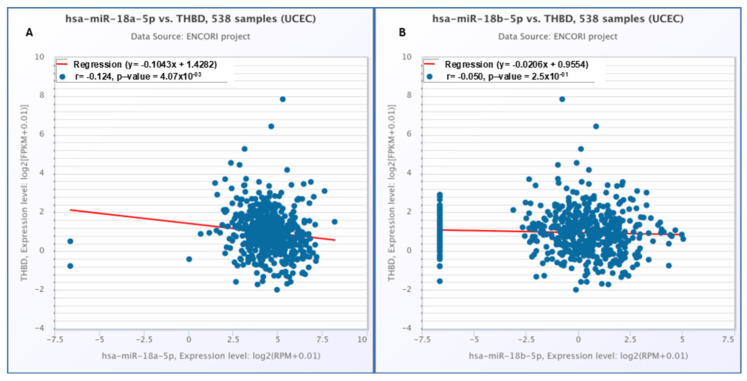
Scatter plots showing the correlation between THBD expression and two microRNAs in UCEC: (**A**) THBD vs. hsa-miR-18a-5p; (**B**) THBD vs. hsa-miR-18b-5p. A significant negative correlation is observed in (**A**), while (**B**) shows no significant relationship. Red line representd correlataion slope and r referes to correlation coefficient.

**Figure 9 ijms-26-03649-f009:**
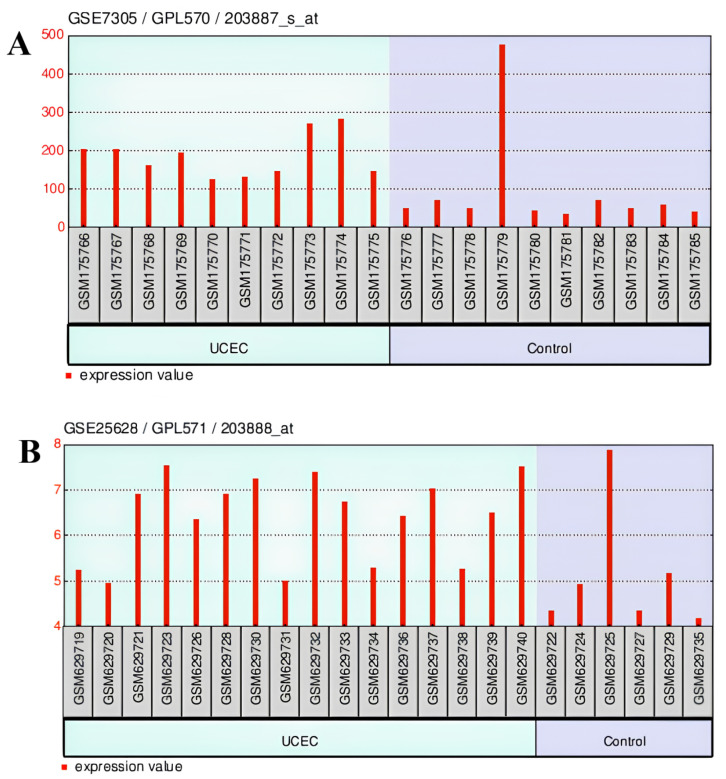
(**A**) The THBD gene change in the GSE7305 dataset, (**B**) the THBD gene change in the GSE25628 dataset (UCEC vs. Control).

**Table 1 ijms-26-03649-t001:** The gene–gene interaction-combined scor result.

Gene-1	Gene-2	Protein Annotation	Combine Scor
*PROCR*	*THBD*	Endothelial protein C receptor	0.999
*F2*	*THBD*	Activation peptide fragment 1	0.999
*PROC*	*THBD*	Vitamin K-dependent protein C heavy chain	0.992
*F3*	*THBD*	Tissue factor	0.982
*F2R*	*THBD*	Proteinase-activated receptor 1	0.966
*TFPI*	*THBD*	Tissue factor pathway inhibitor	0.963
*CPB2*	*THBD*	Carboxypeptidase B2	0.958
*F7*	*THBD*	Coagulation factor VII	0.942
*C3*	*THBD*	Complement C3c alpha’ chain fragment 1	0.933
*PLAT*	*THBD*	Tissue-type plasminogen activator chain A	0.913

## Data Availability

The data generated in the present study may be requested from the corresponding author.

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
