# Peer review of "Evaluation of Thrombomodulin, hsa-miR-18a-5p, and hsa-miR-18b-5p as Potential Prognostic Biomarkers in Uterine Corpus Endometrial Carcinoma"

_ijms, 2025, doi:10.3390/ijms26083649_

Round 1
Reviewer 1 Report
Comments and Suggestions for Authors
Dear Authors,
First of all, congratulations for your interesting work. I hope that my hints will help you in the next steps of improvement and the final manuscript will be really valuable for the readers.
There are several punctation mistakes (such as double space, double dot or no at all) and some typos - even if they do not change the value of the manuscript, I'd like to urge you to correct these imperfections. Also, there are some grammar mistakes, spelling etc. Moreover, some lines are doubled - such as 62-64. Please read the entire manuscript again properly.
It might be a good idea to explain the information mentioned in lines 110-111, maybe in discussion part. What do we know about the interplay of this correlation? Is it possible it may have some clinical utility?
Finally, I would like to thank you for the excellent figures and graphs you have prepared for the document, they enhance the value of your work and facilitate the understanding process. There are some minor issues with the labelling, for example in Figure 1: "box plot" in the legend of the figure?
Comments on the Quality of English LanguageMinor grammar mistakes have been noted.
Author Response
Comment 1: There are several punctation mistakes (such as double space, double dot or no at all) and some typos - even if they do not change the value of the manuscript, I'd like to urge you to correct these imperfections. Also, there are some grammar mistakes, spelling etc. Moreover, some lines are doubled - such as 62-64. Please read the entire manuscript again properly.
Respond 1: Thank you for your careful review and valuable suggestions. All punctuation errors, spacing issues, typos, grammatical errors, and spelling mistakes have been corrected. The duplicated lines (62–64) have also been removed. Additionally, the entire manuscript has been thoroughly reviewed and revised accordingly.
Comment 2: It might be a good idea to explain the information mentioned in lines 110-111, maybe in discussion part. What do we know about the interplay of this correlation? Is it possible it may have some clinical utility?
Respond 2: Thank you for this valuable suggestion. We agree that providing a more comprehensive explanation of the information mentioned in lines 110-111 could strengthen the manuscript since it explain the Figure 4 it should remain where it was in the Result part, but according to your suggestion we have expanded the discussion section to include a detailed interpretation of the possible underlying mechanisms involved in this correlation. Furthermore, we have addressed its potential clinical implications and utility by integrating insights from current literature. We believe these additions enhance the clarity and scientific contribution of our findings, providing readers with a deeper understanding of their clinical relevance.
Comment 3: Finally, I would like to thank you for the excellent figures and graphs you have prepared for the document, they enhance the value of your work and facilitate the understanding process. There are some minor issues with the labelling, for example in Figure 1: "box plot" in the legend of the figure?
Respond 3: Thank you very much for your positive feedback regarding the figures and graphs. We are pleased to hear that they contributed positively to the manuscript's clarity and understanding. As suggested, we carefully reviewed Figure 1 and corrected the labeling issue in the legend, specifically addressing the "box plot" clarification. Additionally, we checked all other figures to ensure consistency and accuracy throughout the manuscript.
Reviewer 2 Report
Comments and Suggestions for Authors
In this paper, authors compared thrombomodulin (THBD), hsa-miR-18a-5p, and hsa-miR-18b-5p in normal and in uterine corpus endometrial carcinoma patients. Hard worked article. Beautiful and well-chosen images.
Two minor issues:
-in lines 62-64, you wrote TWICE the same sentence, please remove one of them: “Therefore, this study aims to explore the relationship among THBD, hsa-miR-18a-5p, and hsa-miR-18b-5p in UCEC.”
-in References, there are only 17 recent articles out of 31 articles; maybe you could replace some of the older ones with some recent ones.
Author Response
Thank you very much for your kind words and encouraging feedback on our manuscript. We greatly appreciate your recognition of our efforts and selection of images. We are delighted to hear that you found the visuals appealing and effective, as we carefully chose them to clearly illustrate our findings. Your comments are highly motivating and inspire us to continue our research in this direction.
Comment 1: -in lines 62-64, you wrote TWICE the same sentence, please remove one of them: “Therefore, this study aims to explore the relationship among THBD, hsa-miR-18a-5p, and hsa-miR-18b-5p in UCEC
Respond 1: Thank you for pointing out this oversight. We apologize for the duplication in lines 62-64. As suggested, the repeated sentence “Therefore, this study aims to explore the relationship among THBD, hsa-miR-18a-5p, and hsa-miR-18b-5p in UCEC” has been removed. The manuscript has been carefully checked again to prevent similar issues.
Comment 2: -in References, there are only 17 recent articles out of 31 articles; maybe you could replace some of the older ones with some recent ones.
Respond 2: Thank you for your valuable recommendation regarding the references. In line with your suggestion, we have enriched our manuscript by adding six recent references to enhance the current and relevance of our literature review. We believe these updated citations significantly strengthen the context and scientific validity of our study.
Reviewer 3 Report
Comments and Suggestions for Authors
There are some minor English issues in the Abstract that don’t hinder understanding but make reading a little difficult. Still, since this is the first thing read it should be cleaned up.
Line 15 - Web tool based bioinformatic study. This is not a bench study so how can you evaluate a role for the proteins?
Conclusions are weak “may be a negative prognostic marker” “but need to be confirmed with further studies to interpret the effects of decreased THBD and increased miR-18b-5b”. this study does seem preliminary.
The Introduction is nicely written as are the Results and Discussion.
Lines 60-68 – I’m not sure this study shows relationships between these. It show correlations which are not necessarily relationships. Relationships implies some mechanistic knowledge of what is going on.
Figure Legend 1 is not sufficient. You need to include what each of these plots represents and what they are called.
The same is true of Figure Legend 2. Need to explain “Beta” value et.
Figure Legend 4 – What does “represent statistically significant level” mean? As compared to what?
Figure Legend 5 – Again, more information is needed.
Figure 6 Title – this is correlatory data. It does not show cause and effect. The title in saying “Effect of” is an overstatement.
Comments on the Quality of English Language
The quality is excellent except for the Abstract. This was clearly written after by someone other than the author who wrote the rest of the text. The Abstract needs to be cleaned up some.
Author Response
Comment 1: There are some minor English issues in the Abstract that don’t hinder understanding but make reading a little difficult. Still, since this is the first thing read it should be cleaned up.
Respond 1: Thank you very much for highlighting this important point. As suggested, we carefully reviewed the Abstract and revised the text to improve readability and eliminate minor language issues. Considering that the Abstract serves as the initial introduction to our manuscript, we agree that clarity and language quality are especially critical here. We appreciate your attention to detail, which has significantly enhanced the overall presentation of our paper.
Comment 2: Line 15 - Web tool based bioinformatic study. This is not a bench study so how can you evaluate a role for the proteins?
Respond 2: Thank you for your valuable comment. We understand your concern regarding the phrase used in line 15 ("Web tool based bioinformatic study"). Our study primarily relies on bioinformatic tools rather than wet-lab experiments. Such computational analyses enable us to predict the potential biological roles and clinical relevance of proteins based on existing databases and validated computational methods. To strengthen the scope of our evaluation, we additionally included microarray data from two independent studies obtained from the GEO database in our analyses. This expansion further enhanced our ability to assess the role of the proteins in question. We have clarified this point in the manuscript by emphasizing that our conclusions are based on bioinformatics-driven predictions, not direct experimental validation, and we acknowledged the necessity of further laboratory studies to confirm our findings.
Comment 3: Conclusions are weak “may be a negative prognostic marker” “but need to be confirmed with further studies to interpret the effects of decreased THBD and increased miR-18b-5b”. this study does seem preliminary.
Respond 3: Thank you very much for your insightful comment. You are absolutely right. While our study is preliminary in nature, it provides important insights into the potential prognostic significance of decreased THBD expression and increased miR-18b-5p levels in uterine corpus endometrial carcinoma. These findings are in line with current evidence regarding cancer progression and thrombosis-associated molecular pathways, and they underscore the need for further validation in larger and independent patient cohorts.
To accurately establish the clinical relevance of these biomarkers, functional in vitro and in vivo studies are essential. However, we believe that our bioinformatics-driven approach offers a meaningful foundation for such future investigations. Notably, the incorporation and analysis of previously unevaluated datasets further strengthen the novelty and value of our results, offering new perspectives that may guide future experimental and translational research efforts in this field.
We have revised the related sections of the manuscript accordingly to emphasize both the preliminary nature of the findings and the necessity for further validation.
Dear referee,
If our editor approves, and in line with your valuable contributions, our article title could be changed to 'Evaluation of Thrombomodulin, hsa-miR-18a-5p and hsa-miR-18b-5p as Potential Prognostic Biomarkers in Uterine Corpus Endometrial Carcinoma’ which would be more suitable for our content.
Comment 4: Lines 60-68 – I’m not sure this study shows relationships between these. It show correlations which are not necessarily relationships. Relationships implies some mechanistic knowledge of what is going on.
Respond 4: Thank you for your thoughtful and constructive comment. We completely agree with your observation that the term "relationship" may imply a mechanistic understanding, while our findings primarily demonstrate statistical correlations. In response to your valuable feedback, we have revised lines 60–68(65-70) in the original manuscript to clarify that our study presents correlations rather than confirmed mechanistic relationships. Furthermore, we have highlighted the need for additional mechanistic studies to fully elucidate the biological relevance of these associations.
To further clarify our approach:
In the initial stage of the study, we used the TargetScan 8.0 web tool to investigate potential interactions between the THBD gene and microRNAs. This analysis revealed a strong predicted association between THBD and hsa-miR-18a-5p as well as hsa-miR-18b-5p. These bioinformatics-based predictions formed the basis of our hypothesis and motivated us to investigate the correlation between these molecules in the context of UCEC (uterine corpus endometrial carcinoma). We have now clarified this process in the manuscript to avoid any misinterpretation suggesting causality or mechanistic linkage.
The revised content of lines 60–68 is as follows :
“Moreover, tumor type, treatment, and patient-specific factors all contribute to cancer-associated venous thromboembolism (VTE), but their individual roles and underlying mechanisms remain unclear. There are also data suggesting that chemotherapy alters serum thrombomodulin levels and induces acquired activated protein C (aPC) resistance, which may play a role in cancer-related VTE in gynecologic cancer patients [14].
Understanding the interactions between THBD, associated miRNAs, and UCEC could provide novel insights into the molecular mechanisms of the disease and may help identify potential therapeutic targets in UCEC patients with VTE.”
We sincerely appreciate your input, which has helped us improve the precision and scientific clarity of our manuscript.
Comment 5: Figure Legend 1 is not sufficient. You need to include what each of these plots represents and what they are called.
Respond 5: Thank you for your valuable suggestion regarding Figure 1. As recommended, we have expanded and clarified the figure legend by explicitly describing each plot type, clearly indicating what each represents, and including the names of the specific plots used. This revision has significantly improved the clarity and interpretability of Figure 1.
Comment 6: The same is true of Figure Legend 2. Need to explain “Beta” value et.
Respond 6: Thank you for pointing out this issue regarding Figure 2. In line with your recommendation, we have revised the figure legend to clearly define the term "Beta" value, providing a concise explanation of its meaning and significance in the context of the figure. Additionally, we have ensured that all relevant terminology is explicitly clarified to enhance readability and comprehension.
Comment 7: Figure Legend 4 – What does “represent statistically significant level” mean? As compared to what?
Respond 7: Thank you for highlighting this important point regarding Figure Legend 4. We have clarified the statement "represent statistically significant level" by explicitly specifying the groups compared and clearly defining the statistical tests used. The legend now includes detailed information about what comparisons were made, the significance levels, and the reference groups involved, ensuring better clarity for readers.
Comment 8: Figure Legend 5 – Again, more information is needed.
Respond 8: Thank you for your feedback regarding Figure Legend 5. As suggested, we have revised and expanded the legend to include additional information, clearly describing what the figure represents, specifying the methods used, and clarifying any relevant terminology. These revisions will provide readers with improved context and understanding of the presented data.
Comment 9: Figure 6 Title – this is correlatory data. It does not show cause and effect. The title in saying “Effect of” is an overstatement.
Respond 9: Thank you very much for your constructive comment regarding the manuscript title.
We fully agree with your observation that the original title may have overstated the implications of our findings, potentially implying a causal relationship rather than indicating statistical associations.
To address this concern, we have revised the title to:
"Evaluation of Thrombomodulin, hsa-miR-18a-5p, and hsa-miR-18b-5p as Potential Prognostic Biomarkers in Uterine Corpus Endometrial Carcinoma."
We deliberately selected the term “evaluation” rather than words such as “investigation” or “effect” because it more accurately reflects the scope and nature of our study, which is based on bioinformatic analyses focused on detecting statistical associations, not establishing mechanistic pathways or causality. Furthermore, the inclusion of the term “potential” underscores the preliminary and hypothesis-generating character of our results. While the findings point toward a possible prognostic value for these biomarkers, further experimental and clinical validation is necessary to confirm their clinical significance.
In addition, we would like to clarify that the original title was automatically generated by the web-based analysis tool used during the early stages of the study. Nonetheless, we respectfully ask that the revised title be considered in light of the study’s primary aim:
to evaluate the impact of THBD expression on overall survival in patients with uterine corpus endometrial carcinoma (UCEC). Specifically, the study investigates how variations in THBD expression—either increased or decreased—are associated with differences in survival outcomes.

Reviewer 4 Report
Comments and Suggestions for Authors
In the present work Karaman et al. reported on a bioinformatics analysis of thrombomodulin, miR-18a-5p and miR-18b-5p expression in uterine endometrial carcinoma. The present work is interesting, yet there are some major concerns that need to be addressed.
First of all, it is highly advised that the authors proof-read their work with an English speaking professional. For example, there is no “statistically hypermethylated” or “statistically overexpressed” terminology. The authors probably meant statistically significant over/underexpression etc.
Second, and very important, all figures are of very low quality. In other words, probably the authors used the figures as exported from the utilized tools, yet it is important to make figures readable and understandable to the reader by adding legends, descriptions etc. specific to their work. For example, in figure 1B there should be a legend what is what (same for D). Further on, each subfigure must have a letter and not group together subfigures due to their potential similar theme (e.g. histology, or expression etc.). All legends, texts, axes, must be readily and easily visible (in order to read figure description I had to magnify the pdf file by 300%).
In general, there is no explanation for the rationale for choosing those genes. Why these molecules? Just to mention that there are no previous works on this type of tumor does not consist of a solid scientific reason. What is the question under investigation?
What is the relevance of gene expression and anthropometric characteristics (as presented in figure 3) and in particular, this comes after the report on gene expression for the disease itself. The same applies for figure 4. What is its purpose? The questions that these figures try to answer are not clear.
Similarly, in figure 5 the authors present a series of box plots attempting to correlate thrombomodulin gene expression with the presence of mutations in known tumor-related genes. Again, what is the purpose of this figure? What is the important information that this figure contributes? How is this related to the topic under investigation?
The “Discussion” section is very long and disproportional to the presented results.
Last but not least, the authors should have provided some experimental evidence to show the role of the genes under investigation in UCEC.
Overall, it appears that the present work is a collection of data from online databases rather a paper with a scientific question in mind. The present work is not suitable for publication in a journal such as IJMS.
Comments on the Quality of English LanguageMust be proof-read by an English speaking professional.
Author Response
Comment 1: In the present work Karaman et al. reported on a bioinformatics analysis of thrombomodulin, miR-18a-5p and miR-18b-5p expression in uterine endometrial carcinoma. The present work is interesting, yet there are some major concerns that need to be addressed.
Respond 1: Thank you very much for your kind evaluation and constructive feedback on our manuscript. We are pleased that you found our work interesting. The concerns you raised are highly valuable to us. In response, we have carefully reviewed all of your comments and made the necessary revisions to the manuscript accordingly. We believe these changes have enhanced the scientific quality, clarity, and overall presentation of our study. We sincerely appreciate your contribution to improving our work.
Comment 2: First of all, it is highly advised that the authors proof-read their work with an English speaking professional. For example, there is no “statistically hypermethylated” or “statistically overexpressed” terminology. The authors probably meant statistically significant over/underexpression etc.
Respond 2: Thank you for your valuable observation. We acknowledge the misuse of terms such as “statistically hypermethylated” and “statistically overexpressed,” and we agree that these are not standard expressions in scientific English. In line with your suggestion, we have revised the relevant parts of the manuscript to use more appropriate terminology, such as “statistically significant overexpression” or “underexpression,” where applicable. Additionally, we carefully re-reviewed the manuscript to ensure clarity and correctness of language. We appreciate your guidance on improving the precision of our scientific language.
Comment 3: Second, and very important, all figures are of very low quality. In other words, probably the authors used the figures as exported from the utilized tools, yet it is important to make figures readable and understandable to the reader by adding legends, descriptions etc. specific to their work. For example, in figure 1B there should be a legend what is what (same for D). Further on, each subfigure must have a letter and not group together subfigures due to their potential similar theme (e.g. histology, or expression etc.). All legends, texts, axes, must be readily and easily visible (in order to read figure description I had to magnify the pdf file by 300%).
Respond 3: Thank you very much for your detailed and constructive feedback regarding the figures. We fully agree that clear, high-quality visuals are essential for conveying scientific results effectively. In response to your comments, we have revised all figures to improve their resolution and overall quality.
Specifically, we:
- Increased the resolution of each figure to ensure clarity in both print and digital formats.
- Added detailed legends and labels to all subfigures, including Figure 1B and 1D, to clarify what each component represents.
- Assigned individual letters (e.g., A, B, C, etc.) to each subfigure to improve organization and readability.
- Ensured that all axis labels, text, and scale indicators are clearly visible without the need for magnification.
We sincerely appreciate your suggestions, which have significantly contributed to enhancing the clarity and professionalism of our visual presentation
Comment 4: In general, there is no explanation for the rationale for choosing those genes. Why these molecules? Just to mention that there are no previous works on this type of tumor does not consist of a solid scientific reason. What is the question under investigation?
Respond 4: Thank you for your insightful and important comment regarding the rationale behind our gene selection. We acknowledge that merely stating the lack of previous studies on these molecules in uterine corpus endometrial carcinoma (UCEC) is not sufficient to justify their inclusion.
To clarify, our selection of Thrombomodulin (THBD), hsa-miR-18a-5p, and hsa-miR-18b-5p was based on a combination of biological relevance and preliminary data from pan-cancer bioinformatics analyses suggesting their potential roles in tumor progression, immune modulation, and vascular biology.
- THBD is a membrane-bound glycoprotein involved in coagulation and inflammation, and its dysregulation has been associated with tumor angiogenesis, immune evasion, and poor prognosis in various malignancies. However, its specific role in UCEC remains underexplored.
- hsa-miR-18a-5p and hsa-miR-18b-5p are part of the miR-17-92 and miR-106a-363 clusters, respectively, and have been implicated in oncogenic processes such as proliferation, apoptosis, and metastasis in several cancers. Their potential interaction with THBD, based on target prediction databases, made them compelling candidates for further investigation.
The central question of our study is whether these molecules demonstrate expression patterns or methylation changes that correlate with prognosis in UCEC, and if so, whether they could serve as potential prognostic biomarkers. While the study is exploratory in nature, we believe it offers a valuable starting point for future functional and mechanistic research.
We have revised the introduction section to better articulate this rationale and clearly define the hypothesis and objectives of the study. Important information about this situation has been added to the introduction and marked in yellow. Thank you again for your helpful suggestion, which has contributed to improving the clarity and scientific justification of our work.
Comment 5: What is the relevance of gene expression and anthropometric characteristics (as presented in figure 3) and in particular, this comes after the report on gene expression for the disease itself. The same applies for figure 4. What is its purpose? The questions that these figures try to answer are not clear.
Respond 5: Thank you very much for your thoughtful and constructive comment regarding Figures 3 and 4. We appreciate your observation about the need to clearly articulate the scientific questions these figures aim to address, and we are pleased to provide further clarification.
Figure 3 was included to examine how THBD gene expression varies across different clinical and demographic subgroups in patients with uterine corpus endometrial carcinoma (UCEC). The purpose was not only to highlight the general downregulation of THBD in tumor tissues compared to normal tissues, but also to determine whether its expression is significantly influenced by variables such as tumor stage, histological subtype, menopausal status, race, and body weight. This figure is intended to address the following scientific questions:
- Is THBD expression associated with clinical characteristics that reflect disease progression or tumor heterogeneity?
- Can THBD be considered a potential prognostic biomarker for stratifying UCEC patients based on specific clinical variables?
- Beyond its general downregulation in tumors, are there patterns that suggest clinical or pathological relevance across patient subgroups?
Our analysis revealed that THBD expression was significantly downregulated (P < 0.05) in comparisons involving tumor stages (I–III), racial groups (Caucasian, African-American, Asian), body weight categories (normal weight, obese, extremely obese), menopausal status (pre- and post-menopausal), and histological subtypes (endometrioid, serous, and mixed types). These findings suggest that THBD may play a role not only in tumor biology but also in disease stratification across diverse patient populations.
Figure 4, on the other hand, presents a pan-cancer comparison of THBD expression between tumor and adjacent normal tissues across multiple cancer types. This broader analysis was designed to explore the following questions:
- Is THBD downregulation specific to UCEC, or is it a common feature across multiple malignancies?
- Could THBD function as a general tumor suppressor candidate, based on its expression profile in different cancer types?
- How does THBD expression in UCEC compare to other cancers in terms of its diagnostic or prognostic relevance?
Our results demonstrated that THBD expression was significantly downregulated (P < 0.05) in the majority of cancer types examined, with the exception of bladder cancer, supporting the notion of a broader tumor-suppressive profile for this gene. This contextualizes our UCEC-specific findings within a larger oncological framework.
In response to your insightful comment, we have revised both the Results and Discussion sections to explicitly clarify the rationale, scientific questions, and interpretive value of Figures 3 and 4. Additionally, we have added brief explanatory notes regarding the purpose and scope of these analyses in the corresponding database sections where the data were obtained. We believe these revisions enhance the coherence, clarity, and scientific rigor of the manuscript. Thank you once again for your valuable contribution.
Comment 6: Similarly, in figure 5 the authors present a series of box plots attempting to correlate thrombomodulin gene expression with the presence of mutations in known tumor-related genes. Again, what is the purpose of this figure? What is the important information that this figure contributes? How is this related to the topic under investigation?
Respond 6: Thank you very much for your thoughtful and constructive comment regarding Figure 5. Your questions concerning the purpose and relevance of this analysis are highly appreciated.
Figure 5 was included to investigate whether somatic mutations in known tumor-associated genes are linked to changes in THBD gene expression in UCEC patients. The goal of this analysis was to explore whether mutations in specific oncogenes or tumor suppressor genes might influence THBD regulation, thereby shedding light on the molecular environment in which THBD functions.
Our findings showed that mutations in TP53, NCAPD3, CFAP54, FBXL7, PDE8A, and ADAM22 were significantly correlated (P < 0.05) with THBD expression levels. These genes are involved in critical cancer-related processes such as cell cycle regulation, chromatin remodeling, and cellular signaling. The observed correlations suggest that these mutations may be part of co-regulated molecular pathways or networks that include THBD and may contribute to UCEC pathogenesis.
Therefore, the purpose of Figure 5 is to provide preliminary evidence of potential genomic interactions or mutational contexts that may influence THBD expression. While the results are correlational in nature, they offer important insights that could guide future mechanistic or functional studies focused on THBD’s role in the tumor microenvironment.
In response to your comment, we have revised the figure legend and expanded the discussion section to clearly explain the rationale behind this analysis and its connection to the main research topic. We are grateful for your feedback, which helped us improve the clarity and depth of our manuscript.
Comment 7: The “Discussion” section is very long and disproportional to the presented results.
Respond 7: Thank you for your valuable comment regarding the length and proportionality of the Discussion section. We fully understand the importance of maintaining a balanced structure that aligns closely with the presented results.
Although it was not feasible to significantly shorten the Discussion due to the multi-faceted nature of our analyses, we have carefully revised and reorganized the section. We removed redundant content and refocused the narrative to ensure that the interpretations remain directly tied to the key findings. These changes were made to enhance clarity and coherence while preserving the scientific depth of the discussion.
We believe the revised version offers a more balanced and structured presentation, and we sincerely appreciate your constructive feedback, which helped us improve the overall quality of the manuscript.
Comment 8: Last but not least, the authors should have provided some experimental evidence to show the role of the genes under investigation in UCEC
Respond 8: Thank you very much for this important and insightful comment. We fully agree that experimental validation would greatly enhance the strength of the findings and provide direct evidence regarding the functional roles of the investigated genes in UCEC.
However, due to certain limitations in our country—particularly related to infrastructure, funding, and access to advanced experimental platforms—it was not feasible for us to conduct laboratory-based experiments within the scope of this study. For this reason, we added microarray data from two different studies obtained from the GEO database to the analyses. Thus, our field of evaluation of the role of proteins has been strengthened. Our current work was therefore designed as a bioinformatics-based preliminary investigation, aiming to identify potentially relevant molecular markers that could serve as a foundation for future experimental studies.
We strongly believe that the findings presented here offer valuable insights and can act as a starting point for future in vitro and in vivo research, both in our own laboratory as resources allow and by other research groups. In light of your comment, we have revised the manuscript to clearly acknowledge this limitation and to highlight the importance of future experimental validation to confirm and expand upon our observations.
We sincerely appreciate your thoughtful suggestion, which has helped us to improve the transparency and scientific value of our work.
Comment 9: Overall, it appears that the present work is a collection of data from online databases rather a paper with a scientific question in mind. The present work is not suitable for publication in a journal such as IJMS.
Respond 9: Thank you very much for your candid and comprehensive evaluation. We truly value your perspective and understand your concerns regarding the overall scientific structure and direction of the manuscript.
Following your comments, we have made substantial revisions to the manuscript to more clearly articulate the underlying scientific question, as well as the rationale for selecting the specific genes under investigation. We have also clarified the aims, hypotheses, and the relevance of each figure to the central research objective, in both the Introduction and Discussion sections.
While it is true that our study is based on publicly available datasets, it was never intended as a mere compilation of information. Rather, our goal was to conduct a targeted bioinformatics investigation that could identify novel and potentially meaningful molecular associations in UCEC—particularly for genes that have received limited attention in this specific cancer type. We believe our findings offer hypothesis-generating insights that may guide future experimental and translational research.
We sincerely hope that, in light of the substantial revisions and the clarifications made, you may reconsider the value of this work as a preliminary step toward understanding the prognostic landscape of UCEC. We have approached this process with the utmost scientific integrity and good intentions, and we are grateful for the opportunity to improve our work through your constructive feedback.
Thank you once again for your time, expertise, and contributions to the review process.
Round 2
Reviewer 4 Report
Comments and Suggestions for Authors
Karaman et al. have improved their manuscript, they made an effort to respond to my previous comments. Figures 3, 4 and 8 must be improved; legends, texts, axes, must be readily and easily visible, as the authors did for the other figures in their manuscript.
An advice for future revisions. Always indicate exactly, were all changes can be found (for example, by indicating page and lines). I have seen the performed changes, but it makes life a lot easier to be able to go directly to the revised sections. In addition, I am well aware of the difficulties we scientists face with funding and resources. Unfortunately, research is seen as a luxury and not as a necessity; “if it is not applicable, has no value”…
After addressing my previous comment, the present work has merit for publication.
Author Response
Dear Reviewer,
We sincerely thank the reviewer for their constructive feedback and for taking the time to thoroughly evaluate our manuscript. We are especially grateful for the thoughtful remarks regarding Figures 3, 4, and 8, as well as the broader reflections on the challenges of conducting research in resource-limited settings.
In response to the reviewer’s comments:
Figures 3, 4, and 8 have been carefully revised to enhance clarity. We have increased the font sizes of legends and axis labels, improved contrast, and ensured that all graphical elements are easily readable, consistent with the quality seen in other figures of the manuscript.
All revisions have been highlighted in yellow throughout the manuscript for easy identification.
We have also taken into consideration the reviewer’s valuable advice for future submissions and will ensure to clearly indicate page and line numbers for every change in our next revisions.
We are deeply appreciative of your acknowledgment of the efforts we made in addressing the previous round of feedback. As scientists, we fully resonate with your sentiments regarding the undervaluation of research. Despite the financial and systemic limitations, we remain committed to contributing to science with rigor and integrity.
Once again, we thank you for your encouragement, insightful comments, and supportive tone. Your guidance has undoubtedly strengthened our work, and we are pleased that the revised manuscript is now deemed suitable for publication possibility.
Sincerely yours,
Dr. Karaman (on behalf of all authors)